# ABCH2 transporter mediates deltamethrin uptake and toxicity in the malaria vector *Anopheles coluzzii*

**Mary Kefi**[1,2]*, **Vasileia Balabanidou**[2], **Chara Sarafoglou**[1,2], **Jason Charamis**[1,2], **Gareth Lycett**[3], **Hilary Ranson**[3], **Giorgos Gouridis**[2], **John Vontas**[2,4]*

**1** Department of Biology, University of Crete, Vassilika Vouton, Heraklion, Greece, **2** Institute of Molecular Biology and Biotechnology, Foundation for Research and Technology-Hellas, Heraklion, Greece, **3** Department of Vector Biology, Liverpool School of Tropical Medicine, Pembroke Place, Liverpool, United Kingdom, **4** Pesticide Science Laboratory, Department of Crop Science, Agricultural University of Athens, Athens, Greece

* mary_kefi@imbb.forth.gr (MK); vontas@imbb.forth.gr (JV)

**Data Availability Statement:** The authors confirm that all data underlying the findings are fully available without restriction. All relevant data are

## Abstract

Contact insecticides are primarily used for the control of *Anopheles* malaria vectors. These chemicals penetrate mosquito legs and other appendages; the first barriers to reaching their neuronal targets. An ATP-Binding Cassette transporter from the H family (ABCH2) is highly expressed in *Anopheles coluzzii* legs, and further induced upon insecticide exposure. RNAi-mediated silencing of the ABCH2 caused a significant increase in deltamethrin mortality compared to control mosquitoes, coincident with a corresponding increase in $^{14}$C-deltamethrin penetration. RT-qPCR analysis and immunolocalization revealed ABCH2 to be mainly localized in the legs and head appendages, and more specifically, the apical part of the epidermis, underneath the cuticle. To unravel the molecular mechanism underlying the role of ABCH2 in modulating pyrethroid toxicity, two hypotheses were investigated: An indirect role, based on the orthology with other insect ABCH transporters involved with lipid transport and deposition of CHC lipids in *Anopheles* legs which may increase cuticle thickness, slowing down the penetration rate of deltamethrin; or the direct pumping of deltamethrin out of the organism. Evaluation of the leg cuticular hydrocarbon (CHC) content showed no affect by ABCH2 silencing, indicating this protein is not associated with the transport of leg CHCs. Homology-based modeling suggested that the ABCH2 half-transporter adopts a physiological homodimeric state, in line with its ability to hydrolyze ATP *in vitro* when expressed on its own in insect cells. Docking analysis revealed a deltamethrin pocket in the homodimeric transporter. Furthermore, deltamethrin-induced ATP hydrolysis in ABCH2-expressing cell membranes, further supports that deltamethrin is indeed an ABCH2 substrate. Overall, our findings pinpoint ABCH2 participating in deltamethrin toxicity regulation.

## Author summary

Malaria control is heavily dependent on insecticides to kill mosquitoes that spread the disease-causing pathogens. Insecticides are absorbed through the legs and sensory head

within the paper and its Supporting Information files.

**Funding:** This study is co-financed by Greece and the European Union (European Social Fund) through the operational programme 'Human Resources Development, Education and Lifelong Learning' in the context of the project 'Strengthening Human Resources Research Potential via Doctorate Research' (MIS-5000432), implemented by the State Scholarships Foundation (IKY) (M.K.). This project has also received funding from "S. Niarchos" Foundation – FORTH Fellowships for PhD candidates within the project ARCHERS: Advancing Young Researchers' Human Capital in Cutting Edge Technologies in the Preservation of Cultural Heritage and the Tackling of Societal Challenges (M.K.). The funders had no role in study design, data collection and analysis, decision to publish, or preparation of the manuscript.

**Competing interests:** The authors have declared that no competing interests exist.

organs when mosquitoes land on treated surfaces, such as bednets or sprayed house walls. The insecticides then must travel through the body to the target site, typically nerve cells, to cause death. In recent years, mosquito resistance to insecticides has become a severe problem leading to increased malaria mortalities, particularly in Africa. Finding out how mosquitoes become resistant is a key step in trying to find a solution to the problem. Previous work showed that some mosquitoes have evolved a resistance mechanism whereby they limit the amount of insecticide that passes through the legs, but how they did this was unclear. Here we show that the mosquito has a transporter, or pump, specifically found in the legs and head organs that actively removes the insecticide back out of the leg, similar to bailing out a boat taking on water, before it can disseminate through the body. The protection provided by the transporter highlights a novel mosquito 'detox' mechanism that could be targeted by insecticidal supplements that block the pump and restore insecticide toxicity.

## Introduction

Malaria is a major impediment to health and prosperity in the Global South [1]. Its prevention is best achieved by vector control which relies heavily on insecticides [2]. Malaria incidence halved between 2000–2015, with the majority of the reduction attributed to the use of insecticides [2]. However, insecticide resistance is a critical threat to vector control, as some mosquito populations now manifest a striking intensity of the resistance phenotype [3].

Due to the mode of insecticide delivery in vector control, *Anopheles* legs are the key sites for contact insecticide uptake, representing the first barrier to be crossed in the insect [4]. Recent studies highlighted the role of mosquito legs in insecticide resistance via structural alterations which reduce the penetration rate of pyrethroids [5] and the overexpression of sensory appendage proteins possibly sequestering the pyrethroid insecticide [6]. An ABCH transporter was also identified in mosquito legs through proteomic and transcriptomic analysis [5,7] and found to be induced by a short-term deltamethrin exposure in *An. coluzzii* [7].

ABC transporters are present in all kingdoms of life functioning as primary-active transporters energized by ATP hydrolysis [8] in the highly conserved nucleotide binding domains (NBDs) that are associated with the translocator transmembrane domains (TMDs) [9]. ABC transporter involvement in insecticide resistance and toxicity has been suggested in some studies [7,10–13], and at the same time a differential expression of subsets of ABC proteins in pyrethroid-resistant *Anopheles* mosquito populations has also been reported [14,15]. Additionally, a number of investigations support the up-regulation of ABC transporters from pests, such as *Bemisia tabaci* and *Plutella xylostella* after exposure to different classes of insecticides, implicating a correlation between pesticide detoxification and transport [16,17]. ABC transporters from the C-subfamily have been implicated in pest resistance to insecticidal pore-forming proteins from *Bacillus thuringiensis* (Bt) [18]. Recently, a multidrug resistance-associated ABC, *Cp*MRP of the polar leaf beetle *Chrysomela populi*, was identified as the first candidate involved in the sequestration of phytochemicals in insects [18,19]. Furthermore, an RNAi toxicology screen in *Drosophila* implicated a C family ABC transporter (CG4562) in spinosad transport, as well as pinpointing the role of the P-glycoprotein orthologue *Mdr*65 as the most important ABC in terms of chemoprotection [20]. Much of this functional analysis of the interaction of ABC transporters with insecticides has been studied through heterologous expression in insect cells and *Xenopus* oocytes [21,22].

Transporters of the H sub-family are present in all insects and other arthropods [23], *Dictyostelium* and zebrafish, but are absent from plants, worms, yeast, or mammalian genomes [24]. The H sub-family transporter members are export proteins sharing similarities with members of the G group [9]. They are composed of a single NBD and TMD, hence they are half-transporters, which means they need to dimerize to be functional [25,26]. In *An. coluzzii*, three genes encode ABCH transporters (*ABCH1*, *ABCH2* and *ABCH3*) [26]. ABCH transporters have been implicated in lipid transport, as knock-down experiments in *Drosophila melanogaster*, *Tribolium castaneum*, *Locusta migratoria*, *Plutella xylostella* and *Nezzara viridula* result in high lethality due to desiccation [27–31]. Immunofluorescence analysis showed that the *Drosophila Snu* protein, an ABCH2 orthologue, localizes to the apical plasma membrane of the epidermal cells of larvae.

ABCH transporters have been found differentially expressed in insecticide resistant populations or after insecticide exposure of several insect species [7,11,32–36], In *Anopheles*, all three ABCH gene transcripts are enriched in sensory appendages (ABCH2 and ABCH1 in antennae and palps, while ABCH3 only in palps) [15,37]. In the legs, proteomic analysis could only detect the ABCH2 transporter, whereas more sensitive transcriptomic studies indicated differential regulation of the 3 H class transporters in the legs (as well as G, C, E subfamily members) after short-term deltamethrin exposure [7]. This study showed that ABCH1 and ABCH2 were up-regulated, while ABCH3 was down-regulated post-deltamethrin exposure [7]. However, in a separate transcriptomic dataset taken from adult *An. coluzzii*, ABCH2 was found to be down-regulated in the whole body extracts, 12 hours after sublethal deltamethrin exposure [10], potentially indicating a tissue specific response to insecticides of ABCH2 expression following insecticide exposure.

To initiate the study of the ABCH family we have focused on differentially regulated ABCH2 gene. We were interested to determine whether its upregulation specifically in the legs following insecticide exposure was linked to the toxicity of deltamethrin and if so was this due to direct binding/transport of insecticides or a consequence of other physiological roles.

## Results

### RNAi-mediated silencing of *ABCH2* revealed its implication in pyrethroid toxicity

As described above, former work of our group showed that *ABCH2* was induced upon deltamethrin exposure in mosquito legs [7] and this was confirmed by RT-qPCT expression analysis (S1 Fig). Moreover, ABCH2 was previously identified uniquely in the leg proteome of *An. coluzzii* [5] and as well in the leg transcriptome [7]. Here, more precise tissue specific analysis of expression was performed, both at the transcript and protein level. The transcript abundance in different dissected body parts and tissues of 3–5 day-old female *An. coluzzii* was evaluated with RT-qPCR, with the expression of all samples being normalized against the expression of abdominal walls. As expected, the *ABCH2* relative expression in legs is greater than other dissected tissues (S2A Fig). The protein abundance of ABCH2 is higher in the legs with protein detected in heads too, as evidenced by western blot analysis (S2B Fig). The ABCH2 expression in head is detected in specific head appendages (antennae, proboscis, maxillary palps) (S2C Fig) and not to the rest of the head.

To test whether *ABCH2* exhibits phenotypes related to pyrethroid toxicity, we performed RNAi-mediated silencing in newly emerged adults of a recently colonized *An. gambiae* population from Burkina Faso [38,39], followed by bioassays. dsRNA specifically targeting *ABCH2* transcripts were designed, generated and introduced intrathoracically into newly emerged *An. coluzzii* females via nano-injections. The silencing efficiency was evaluated both at the

transcript and protein levels. According to RT-qPCR, dsRNA-mediated silencing reduced *ABCH2* transcript levels in whole female mosquitoes by approximately 80% (Fig 1A). Western blot analysis using the ABCH2 peptide antibody detected a specific signal at approximately 85 kDa, the expected size of the transporter and was used to prove the silencing efficiency at the protein level. Indeed, ABCH2 protein in legs and head appendages (proboscis, antennae, and maxillary palps) of female mosquitoes were barely detectable compared to ds*GFP* treated counterparts (Fig 1B). Due to sequence similarity of ABCH2 with the other ABCH transporters, we also tested potential non-specific targeting of the dsRNA construct against them and find it non-significant (S3 Fig).

Next, the effects of *ABCH2* silencing in pyrethroid toxicity were evaluated via deltamethrin toxicity assays. Mosquito knock-down 1 hour post exposure in ds*GFP* injected mosquitoes was 23% while in dsA*BCH2* mosquitoes, it was 89.7%, showing a significant increase (two-sided Fisher's exact test, *p*-value<0.0001). The ds*ABCH2* mosquitoes did not recover, with a mortality of 98% after 24 hours, while in ds*GFP* controls, the mortality was 55% (two-sided Fisher's exact test, p-value<0.0001) (Figs 1C, 1D and S4).

To rationalize the increased deltamethrin mortality, we determined the amount of internalized deltamethrin by using $^{14}$C-deltamethrin contact toxicity assays on silenced and control *An. coluzzii*. ds*ABCH2* knockdown mosquitoes exhibit increased $^{14}$C-deltamethrin penetration compared to ds*GFP* controls by about 10% (Paired t-test, p-value = 0.0037) in the three biological replicates (Fig 1E and S2 Table).

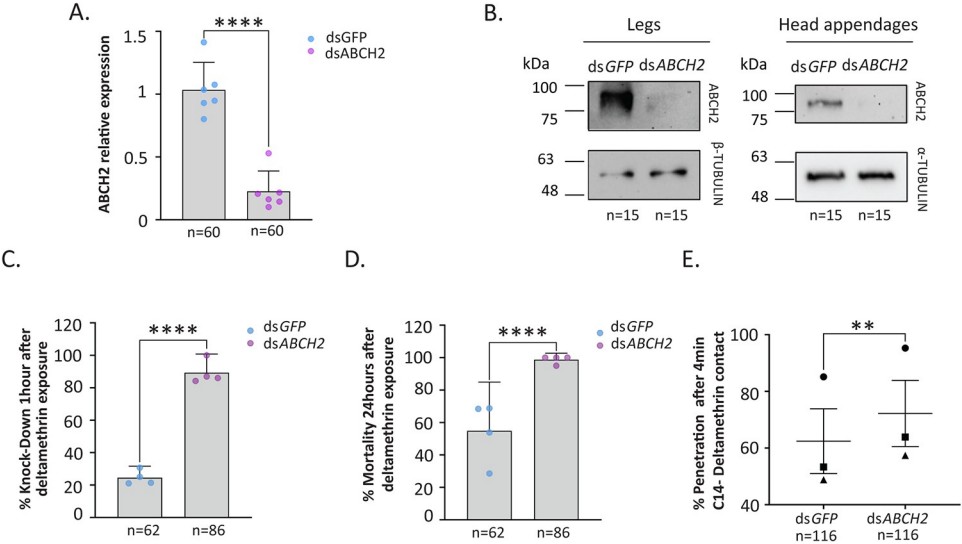

**Fig 1. RNAi silencing efficiency and deltamethrin toxicity assays.** A. Silencing efficiency estimation of ds*ABCH2* against ds*GFP* injected females with RT-qPCR. ABCH2 transcript level reduction by 78% accomplished, as indicated by the mean of 6 biological replicates (depicted with semi-transparent circles), with 95% confidence intervals (CI); *P*-value <0.0001 (****) determined by unpaired t-test, B. Western Blot analysis of leg and head appendages protein extracts 3 days post injections, verified the reduced protein levels of ds*ABCH2* against ds*GFP* injected controls. Alpha or beta-TUBULIN were used as loading control, C. % Knock down of ds*ABCH2* and ds*GFP* female *An. coluzzii*, subjected, 72h post injection, to 1 hour deltamethrin exposure (0.016%); Mean$_{(dsGFP)}$ = 23% + 95% CI and Mean$_{(dsABCH2)}$ = 89.6% +95% CI for n = 4 biological replicates depicted with transparent circles; *P*-value<0.0001 (****) determined with two-sided Fisher's exact test, D. % mortality 24 hours post exposure, Mean$_{(dsGFP)}$ = 55% + 95% CI and Mean$_{(dsABCH2)}$ = 98.75% +95% CI, for n = 4 biological replicates; *P*-value<0.0001 (****) determined with two-sided Fisher's exact test, E. % penetration of $^{14}$C-deltamethrin after 4 minutes of contact in ds*GFP* and ds*ABCH2* mosquitoes. Penetration corresponds to the percentage of internal counts compared to total counts (external and internal) at this time point. Mean+/- SEM of three biological replicates (n = 116 total mosquitoes/ condition). Replicate pairs are presented with different shapes (Rep 1: triangle, Rep2: square, Rep3: circle). Mean difference is 9.77; *P*-value = 0.0037(**), determined with two-tailed paired t-test (S2 Table).

## ABCH2 appears localized in the leg/appendage epidermis, underneath the cuticle, with apical polarity

As ABCH2, expressed in leg and head appendages, was thus implicated in deltamethrin toxicity, our next objective was to determine its (sub)cellular localization in these tissues to gain more insights into its role in insecticide toxicity. Towards this, a specific antibody against this transporter was used in immunofluorescence experiments in leg cryo-sections. We consistently obtained a specific signal (green) underneath the cuticle, as shown in the merged-fluorescent images with the bright-field channel (Fig 2A and 2B). The signal observed followed the linear contour of the leg, just underneath the cuticle and adjacent to the nuclei stained red with TO-PRO-3 dye. Also stained were characteristic triangle-shaped protrusions towards the cuticle. To confirm this location as sub-cuticular epithelia, we obtained a similar staining pattern on leg cryo-sections with an E-cadherin antibody, known to stain these cells [40–43] (Fig 2C and 2D). The ABCH2 signal appeared to be located apically, as opposed to the more even distribution of Cadherin over the whole plasma membrane.

A similar ABCH2 staining pattern was also observed in cryo-sections of head appendages (Fig 2E–2H). Furthermore, using the DeepLoc-1.0 tool [44], we predicted a plasma membrane

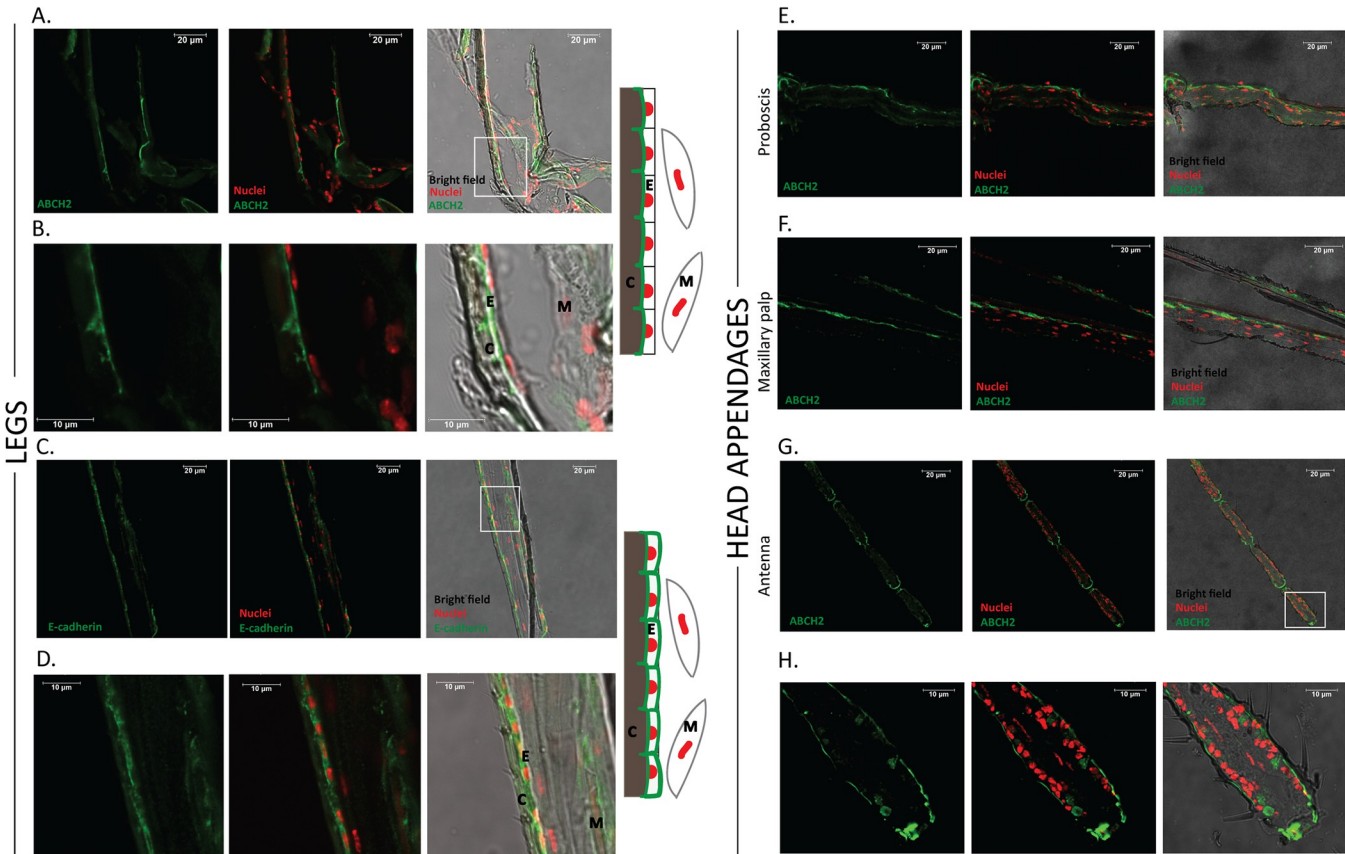

**Fig 2. Sub-cellular localization of ABCH2 transporter.** Immunohistochemical images from longitudinal leg cryosections of 3–5 day old female *An. coluzzii*. A. ABCH2 localization on epidermal cells, underneath the cuticle, polarized towards the apical side. C. Epithelial staining on leg cryosections with a marker-antibody against E-cadherin validating the presence of an epidermal layer underneath the cuticle. B, D. Zoomed images of the selected, white squares of figures A and C respectively, together with graphical depictions of the main structures observed. E,F,G. ABCH2 localization on epidermal cells of head appendages: proboscis, maxillary palp and antenna respectively. H. Zoomed image of the selected, white square of figure G. Images were obtained with confocal microscope (40x). RED: Cell nuclei are stained with TO-PRO-3; Green: antibody staining; Merged images with and without bright-field channel are also depicted; C = cuticle, E = epidermis, M = muscles; scale bars of 10 or 20 μm are illustrated.

sub-cellular localization for this multi-span transmembrane protein (S5 Fig). These lines of evidence support that mosquito ABCH2 is localized on leg/appendage epidermal cells, most probably apically polarized towards the cuticular structure.

## Deltamethrin toxicity could not be attributed to CHC differences in the legs

As several studies implicate ABCH transporters in lipid transport, we next sought to test whether the increased mortality and penetration in ABCH2 depleted mosquitoes is due to reduction of lipid species. *An. coluzzii*, *D. melanogaster*, *L. migratoria*, *P. xylostells* and *T. castaneum* possess three *ABCHs* (with one-to-one orthology relationships). Interestingly, all ABCHs with reported roles in lipid transport are clustered within the same clade with AcABCH2 (S6 Fig). Thus, we wondered whether ABCH2 also participates in transport of cuticular hydrocarbons (CHCs), being the most abundant lipid species in *Anopheles* leg cuticle [45] and having a documented role in reducing the penetration rate of insecticides [45]. To address this, we performed CHC analysis in mosquito legs from control or ds*ABCH2* injected mosquitoes. *ABCH2* silencing of the injected mosquito batch was verified with western blot analysis (S7 Fig). The analysis of hexane leg extracts indicated no significant difference either in the total CHC content between ds*ABCH2* and ds*GFP* mosquito legs (Fig 3), or in the relative abundance of individual CHC species (S8 Fig). These results suggest that the increase in

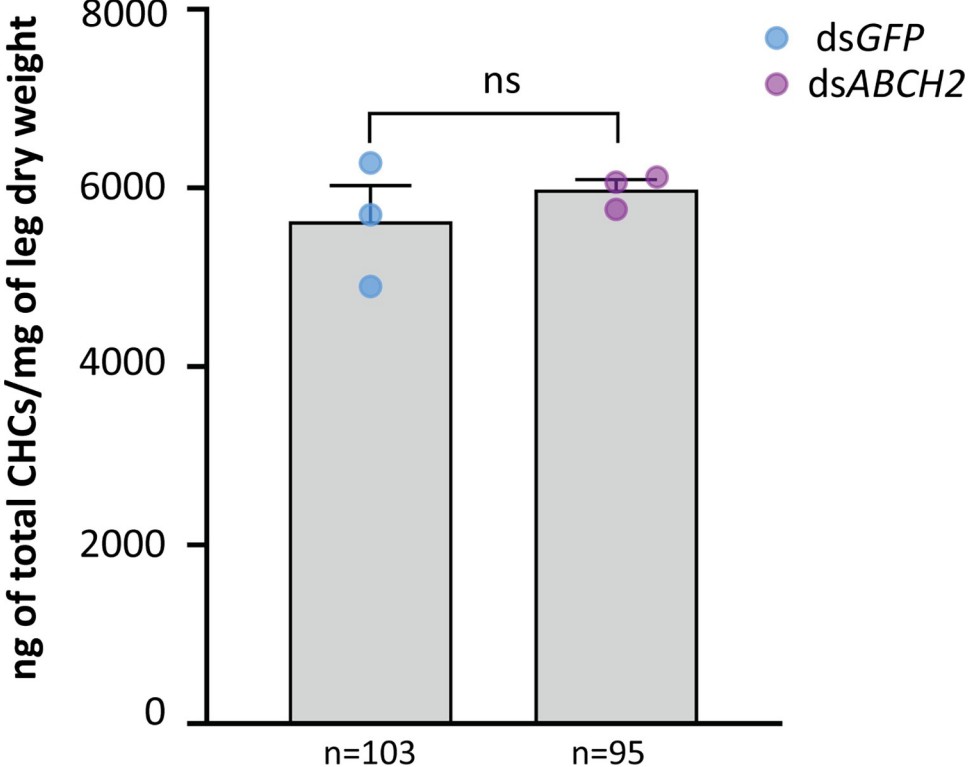

**Fig 3. Total Cuticular Hydrocarbon content of legs derived from ds*ABCH2* and ds*GFP* injected mosquitoes.**
Average total CHC content from three replicates analyzed for each condition (95 ds*ABCH2* and 103 ds*GFP* female mosquitoes, 30–35 mosquitoes per replicate); Mean (ngCHC/mg dry leg weight) + SEM; Mean$_{(dsGFP)}$ = 5626.33+ 401.5 and Mean$_{(dsABCH2)}$ = 5981.67+ 112.5 for n = 3 biological replicates, depicted with semi-transparent circles; *P*-value = 0.4422 (non-significant, ns), determined with unpaired t-test.

deltamethrin toxicity observed in the ABCH2-silenced mosquitoes is not due to a change in the amount or profile of epicuticular hydrocarbons.

### *In silico* and *in vitro* tools provide evidence for a membrane-bound ABC transporter which functions most probably as a homodimer

The ABCH2 sequence was modelled after the homodimeric ABCG1 structure [46], identified to be its closest homologue with an available structure. This analysis identified contacts stabilizing the inter-protomer interfaces using the Protein Interaction Calculator (PIC) web-server [47]. The number and nature of interactions stabilizing the dimeric ABCG1 interface is equivalent to those occurring in the modelled ABCH2 one (Fig 4A and 4B and S1 File), suggesting that ABCH2 adopts a homodimeric state. To verify this, we expressed ABCH2 using the baculovirus system and assessed its ATP hydrolytic activity *in vitro*. After validating the specific expression in ABCH2-infected cells compared to cells infected with an empty-bacmid (S9A Fig), we proceeded to sub-cellular fractionation. As expected, the transporter is only present in the membrane fractions of the ABCH2-infected cells (Figs 4C and S9A, S9B). Inverted Membrane Vesicles (IMVs) isolated from these cells, indicate an ATP hydrolysis specific to ABCH2 expression, as the amount of Pi released was increased significant compared to IMVs expressing an unrelated membrane protein (Figs 4D and S10). In order to approximate the rate of ABCH2-related ATP hydrolysis, we performed comparative western blot analysis with known quantities of a reference purified His-tagged protein, to estimate the amount of His-ABCH2 in expressing IMVs (S11 Fig, S3 Table). Based on this estimate, free ABCH2 hydrolyzes ~178pmoles Pi/pmol ABCH2/minute (S4 Table), representing a substantial basal hydrolysis rate compared to other membrane-embedded motor proteins [48,49].

### Docking and *in vitro* experiments indicate that deltamethrin is a putative substrate of ABCH2

To validate whether deltamethrin can act as an ABCH2 substrate, we initially performed docking analysis *in silico*. The model used ABCG1 which has been crystallized in many distinct cholesterol-liganded states [46]. In the nucleotide-free inward facing conformation (PDB:7R8D), cholesterol molecules have been allocated to the transmembrane part of the transporter. Two of them at the channel interior and three additional ones at its exterior. In the ATP bound state (PDB:7R8E), the cholesterol molecules were observed solely in the channel exterior. We assessed the cholesterol binding pockets via the Protein-Ligand Interaction Profiler Web-server and focused our subsequent analysis on the most well defined, interior pockets, present within the translocation path [46]. Initial protein-ligand docking experiments were carried out using AutoDock Smina [50] to retrieve the binding energies of cholesterol and deltamethrin. From the top scores, we show (Fig 5A and 5B) the ones having binding modes resembling closely the crystal structure [46]. The results indicate that both ligands docked to the pocket with similar energies (Fig 5A). Moreover, the number and nature of interaction stabilizing the two ligands into the pocket is equivalent (Fig 5A). These observations suggest deltamethrin acting as a potential substrate of ABCG1. Next, to assess whether deltamethrin is an ABCH2 substrate, we performed protein-ligand docking experiments on the modeled ABCH2 without a bias related to the substrate pocket. Thus, we widened our search grid to include the entire translocator (i.e. transmembrane region). The top scores that emerged indicate a substrate pocket in the translocator interior, just above the one defined by the crystal structure (compare Fig 5A *vs* 5B, left panels). In such a pocket, both cholesterol and deltamethrin dock with equivalent orientation, energies, and interactions (Fig 5B). To look for experimental evidence of binding, deltamethrin was introduced in the *in vitro* assay to ask whether it stimulates the

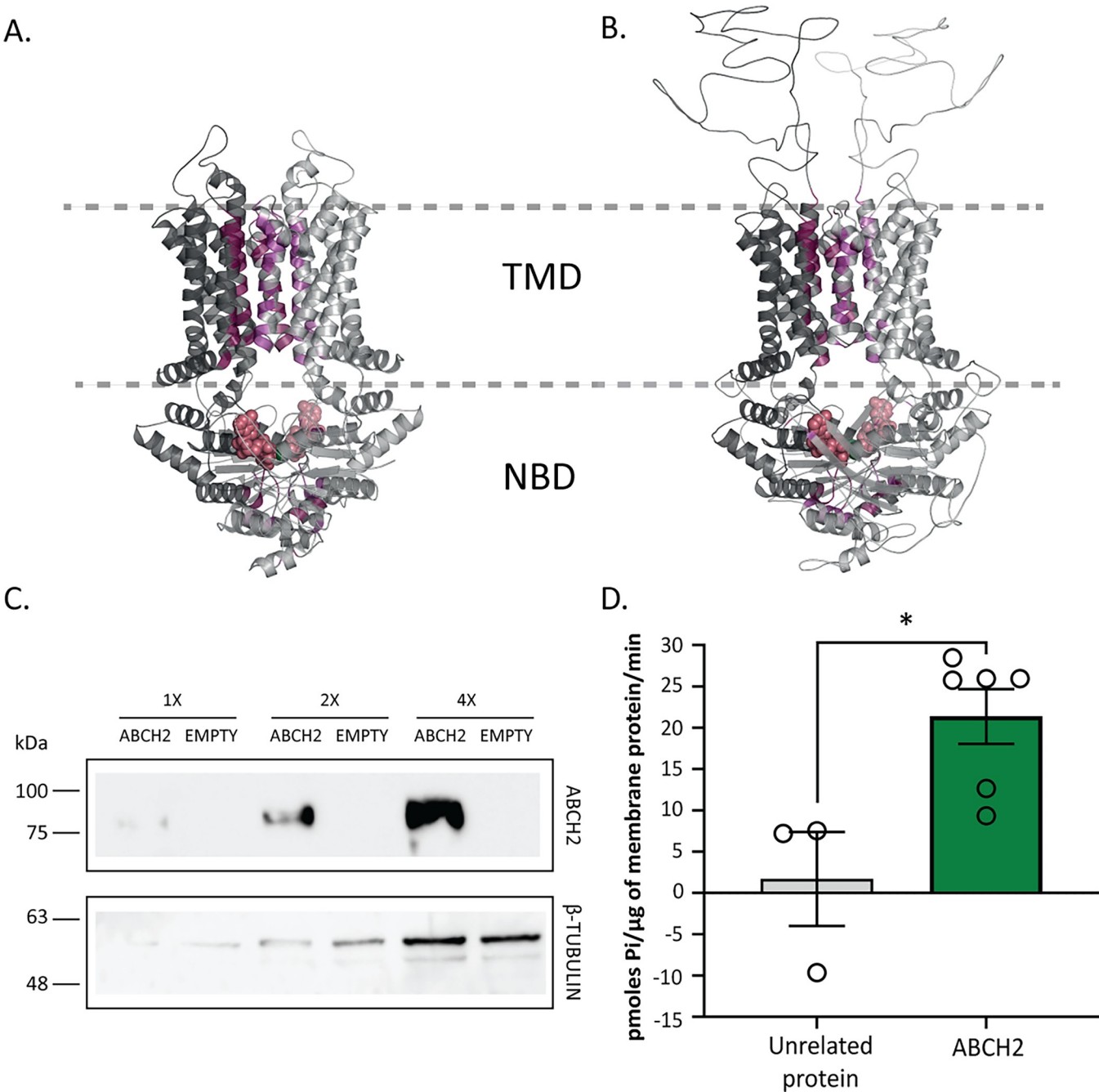

**Fig 4. Functional expression of ABCH2, as a homodimer.** A and B. Interface residues participating in interactions stabilizing the homo-dimeric interface are presented with purple surface colors. Bound ATPs in the Nucleotide Binding Domains (NBDs) are presented with red spheres. The two protomers in the ABCG1 structure (A) or in the modelled ABCH2 (B) are distinguished by two grey scales. C. Western blot analysis using ABCH2 antibody in whole membrane preparations of ABCH2- and Empty Bacmid-expressing *Sf9* cells. Different concentrations (1x, 2x, 4x) of membrane preparations were tested with B-TUBULIN serving as loading control. D. ATPase activity of membrane preparations using malachite green. Mean + /- SEM of six biological replicates in ABCH2- membranes and 3 biological replicates in unrelated protein-expressing membranes and the average of 2–3 technical replicates for each biological is presented in bars. Mean pmoles Pi/μg protein +/- SEM; Mean$_{(EMPTY)}$ = 1.7+/- 5.67 and Mean$_{(ABCH2)}$ = 21.37+/- 3.32; *P*-value = 0.0148 (*).

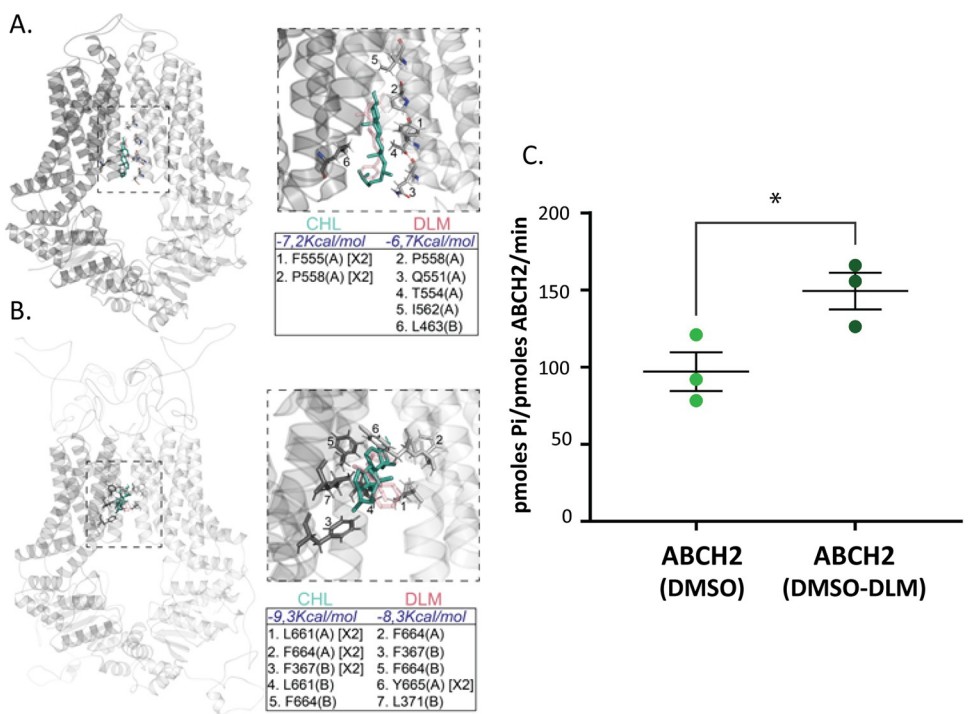

**Fig 5.** A. ABCG1 having docked cholesterol (CHL) and deltamethrin (DLM) as described in the text and methods. CHL is presented with green sticks, whereas DHL with transparent red sticks. Zoom-in of the indicated dotted area is presented (right top panel) together with the calculated energies and substrate-stabilizing interactions (right bottom panel). The residues participating in the hydrophobic interactions with the substrates are given in the table for each protomer (A) or (B), and the number of hydrophobic bonds for each residue is indicated in brackets (when more than one), B. As in A with the modelled ABCH2 structure, C. ATPase activity of membrane preparations using deltamethrin. Pmoles of Pi produced per pmole of ABCH2 per minute of reaction are depicted in ABCH2-expressing IMVs in reaction containing the organic solvent (DMSO) and deltamethrin (DMSO-DLM). Mean pmole Pi/pmole ABCH2/min +/-SEM of three biological replicates performed in duplicate are depicted. Mean$_{(DMSO)}$ = 97.07+/-12.5 and Mean$_{(DMSO-DLM)}$ = 149.3 +/- 11.9; *P*-value = 0.0394 (*), determined with unpaired t-test.

basal ABCH2-IMVs ATPase activity. Interestingly, a significant stimulation over the basal activity was observed when deltamethrin was included in the reaction (Fig 5C and S5 Table). Taking all together, we propose deltamethrin to be a physiologically relevant ABCH2 substrate.

## Discussion

Mosquito legs/appendages are the first line of defense against insecticides when *Anopheles* vectors come in contact with impregnated bed-nets and wall surfaces [4,51]. The presence of transport systems in appendages apart from serving key physiological functions, could be involved in conferring tolerance against chemicals, such as insecticides [15]. But whether and how mosquito transporters are implicated in drug toxicity remains largely elusive. Here, we show that the ABCH2 transporter has an essential role in alleviating pyrethroid toxicity in *An. colluzii*, presumably by direct transport of pyrethroids out of cuticular epithelial cells, based on several lines of evidence.

Firstly, RNAi-mediated silencing of the *ABCH2*, an ABC transporter previously shown to be induced by deltamethrin, increased pyrethroid toxicity substantially (from 23% to 90% knocked-down mosquitoes and from 55 to almost 99% mortality) (Fig 1C and 1D). Moreover, [14]C-Deltamethrin penetration experiments revealed that *An. coluzzii* lacking the transporter

showed increased penetration of radiolabeled insecticide after short (4 min) contact (Fig 1E). It is plausible that pumping out the insecticide, gives greater time for metabolic enzymes to detoxify the insecticide, making the transporter-devoid mosquitoes far more vulnerable to insecticides.

Secondly, the anatomical localization, in relevance to insecticide uptake by legs and appendages and the cellular localization within epidermal cells with an apical polarity, oriented towards the cuticle (Fig 2) [43], is consistent with the ability of the transporter to remove insecticides from the interior of the organism at the first point of entry. The ABCH2 orthologue in *D. melanogaster* is also localized in epidermal cells, indicated by epidermal-Gal4/RNAi screening and by localization experiments with fused GFP, showing its topology on the apical surface of larval epidermal cells [29]. It is common for transporters to compartmentalize on the membranes towards the site they exhibit their function [52]. A polarity towards the outermost part, i.e. the exoskeleton suggests the ability of the transporter to facilitate transport of cuticular components and/or other substrates, such as insecticides, out of the organism and both scenarios were explored.

Thirdly, we tested the scenario of transport of deltamethrin across cell membranes that would lend support to the difference in penetration in being attributed to direct export from epidermal cells and sequestration of the insecticide into the cuticular structure. Several reports have shown that ABC transporters can participate in phase 0 detoxification [53]. To explore this possibility, we performed *in silico* analysis and functional ABCH2 expression *in vitro* which revealed that ABCH2 is a functional transporter of homodimeric nature. ABCHs are half-transporters, meaning they need to form either homo- or hetero-dimers in order to be functional [26]. *In silico* data support the homodimerization scenario due to the similar nature and number of interface interactions observed in the modelled ABCH2, as in the homodimeric ABCG1 crystal structure (Fig 4A and 4B and S1 File). Additionally, the elevated ATP hydrolysis observed in the ABCH2-IMVs in the absence of a substrate (Fig 4C), is indicative of an active transporter with significant basal activity, as previously reported for other ABC transporters [49]. Membrane preparations from recombinant *Baculovirus* infected Sf9 cells have been widely used to detect interactions of compounds with ABC transporters of several families [54–56], however to our knowledge this is the first *in vitro* functional expression of an ABC transporter of the H sub-family. Moreover, evidence of insect ABCH2 orthologues also supports the homodimer role in resistance, since in RNAi screens [of *D. melanogaster* [29,57], *N. viridula* [30], *T. castaneum* [27]] mortality was observed only with ABCH2-orthologues and no other paralogues exhibited similar phenotypes (mortality).

Our ensuing objective was to test whether deltamethrin is an ABCH2 substrate. In general, providing direct proof of the ability of a transporter to translocate a specific molecule is challenging [58]. Decades of research in biological transport systems have highlighted the difficulty in assigning a substrate to an ABC transporter and the low reproducibility arising when using different systems [58]. As a first indication that deltamethrin could be a substrate of ABCH2, we performed protein-ligand docking analysis on the modeled ABCH2 structure and demonstrated that, deltamethrin could act as an ABCH2 ligand by the identification of a putative binding pocket (Fig 5A and 5B). To further substantiate that deltamethrin is indeed an ABCH2 substrate, we performed ATP hydrolysis measurements on isolated membrane vesicles and we observed a substantial stimulation over its basal activity in the presence of deltamethrin. Based on *in silico* and *in vitro* evidence, we thus tentatively conclude that deltamethrin is indeed an ABCH2 substrate (Fig 5C).

Lastly, we explored the possibility that the reduced insecticide penetration could be associated with the amount of cuticular lipids on ABCH2-silenced mosquitoes. Based on our phylogenetic analysis, *An. coluzzii* ABCH2 is an orthologue of other insect ABCH transporters

implicated in transport of cuticular components in early developmental stages [30]. In the insect, knockdown studies have associated ABCH transporters with cuticular lipid transport abnormalities primarily through lipid staining of silenced individuals. Albeit, to our knowledge, the exact molecules transported by this essential protein have not yet been identified [27–29,31]. Recent work on another ABCH transporter, oksyddad (orthologue to ABCH1, S6 Fig), indicates that it is required for CHC deposition at the surface of the wing cuticle either directly or indirectly, while it does not directly participate to the *D. melanogaster* cuticular barrier formation, shown to be mediated by the ABCH2-orhologue Snu. [57]. Another study revealed that the leg epicuticle of insecticide resistant *Anopheles* mosquitoes, which uptake deltamethrin slower upon tarsal contact, is thicker mainly due to enhanced deposition of cuticular hydrocarbons, the major lipid species in leg epicuticles [5]. When we analyzed *ABCH2*KD legs versus controls, however, the total hydrocarbon content in both cases were similar and the relative abundances of each CHC species identified also showed non-significant differences (Fig 3). Therefore, the increase in deltamethrin toxicity demonstrated in the *ABCH2*-silenced mosquitoes is not attributed to the reduction of CHCs, at least in the time frame we tested.

Summarizing, several lines of evidence presented here support that deltamethrin could be an ABCH2 ligand (and a putative substrate for translocation): 1) the significant increase in mortality and insecticide penetration in early time points in ABCH2-silenced *An. coluzzii*, 2) the localization of the transporter on the apical membrane of the epidermal layer is in accordance with the hypothesis that this molecule is able to pump out compounds into the exoskeleton and perhaps out of the organism, 3) the *in silico* docking experiments on the modelled ABCH2, derived by the available crystal structure of the ABCG1 homolog, indicates that deltamethrin is a potential ABCH2 substrate, 4) the increased deltamethrin-induced ATP hydrolysis in the *in vitro* system. Based on the above, we propose that ABCH2 mediates deltamethrin efflux in *An. coluzzii*.

Although eukaryotic ABC transporters are well-known as efflux pumps, in the absence of direct export evidence we cannot exclude alternative mechanisms of ABCH2 action that would lead to higher mortality after silencing. One possible scenario would be ABCH2-mediated deltamethrin sensing and triggering of signaling pathways that lead to slower insecticide entrance or stimulation of the detoxification process. Such a mechanism has been described so far in bacterial systems and specifically in *B. subtilis*, where BceAB participates in bacitracin resistance as a co-sensor for signal transduction [59,60]. Future studies focusing on the development of membrane reconstitution systems, that is purified proteins incorporated into artificial lipid membranes (liposomes) [61,62], may provide direct evidence regarding deltamethrin translocation.

Additionally, based on our datasets, we cannot fully exclude other possibilities concerning the toxicity response we observe. Our CHC analysis in ABCH2-silenced versus control mosquitoes did not show any significant differences either in total CHC contents nor in any of the 32 individual CHC species identified. We focused on CHCs as it is the major lipid species in *Anopheles* legs, however other (epi)cuticular lipids, such as fatty acids, esters, alcohols, sterols, phenols or other components of the envelope such as wax and cement of unknown composition [63] not tested here, could be ABCH2 substrates, with relevant implication in the observed phenotype.

Our data support that ABCH2 is involved in an acute insecticide toxicity response rather than an adaptive mechanism constitutively expressed in insecticide resistant populations. One possible explanation for that is that ABC transporter overexpression might be costly for the mosquito, especially in the absence of the substrate that stimulates this response (in that case deltamethrin). This is in line with the fact that several ABC-transporters are differentially expressed in multiple datasets upon insecticide exposure [10–13,64]. This was also the case in

*An. coluzzii* legs transcriptome analysis where a plethora of ABCs was identified after short-deltamethrin exposure rather than found constitutively upregulated in resistant mosquitoes [7]. There were also no constitutively expressed ABC transporters found differentially regulated in in *An. coluzzii* leg proteome from resistant mosquitoes either [5]. This might be another indication that ABCH2 participates in a fast-acting mechanism present in the insecticide entrance point, beneficial for the mosquito, as it would give some time to allow detoxification responses to occur.

Altogether, our results provide evidence for an ABC transporter-based, fast-acting resistance mechanism present at the appendages. As mosquito legs/appendages are a highly epidemiologically relevant tissue with vector control relying on insecticide penetration through the legs, the association of transporter proteins with pyrethroid toxicity and the understanding of the underlying molecular mechanisms would be very useful in vector control innovations. Due to the druggable characteristics of the ABCH2 molecule, specifically, an arthropod-specific, plasma membrane-bound transporter present in the cuticular epidermis of mosquito appendages, we consider this as a potential target for insecticide formulation add-ons to synergize pyrethroid toxicity.

## Materials and methods

### Mosquito strains

The mosquito strain used in this study belong to the *An. coluzzii* species complex and was maintained in the laboratory under the same conditions for several generations before analysis. The standard insectary conditions for all strains were 27°C and 70–80% humidity under a 12-h: 12-h photoperiod with a 1-h dawn:dusk cycle. The strain is derived from Burkina Faso (VK7) [38, 39] and the colony used for the experiment is the VK7-LR colony, which has lost part of its resistance, as described in [7].

### Antibodies

Rabbit polyclonal antibodies targeting *An. coluzzii* ABCH2 peptide (residues 466–482: VKEYYSDLDSALGAVRD) were synthesized and affinity purified by Davids Biotechnologie. E-cadherin mouse antibody was kindly provided by Dr. Siden-Kiamos [65]. Mouse anti-beta-tubulin and mouse anti-alpha-tubulin used for normalization control in western blot analysis were purchased by Santa Cruz Biotechnology and Developmental Studies Hybridoma Bank respectively.

### Western blot analysis

To determine *ABCH2* relative protein abundance in different tissues, 3–5 day-old female *An. coluzzii* were dissected, RNA-extracted and subjected to western blot analysis using an antibody against ABCH2 and anti-tubulin serving as loading control. Dissected tissues (abdominal walls, legs, heads, midguts/malp.tubules/ovaries) from 10–15 female mosquitoes were homogenized into 100–150 µl RIPA Buffer (50Mm Tris pH = 8, 150mM NaCl, 1% SDS, 1% NP-40, 1mM EDTA, 1mM EGTA and 1 mM PMSF). After a 10 min centrifugation step at 10,000g the supernatants were immediately supplemented with equal volume (100–150µl) of Laemmli Sample Buffer. Polypeptides were resolved by 12% acrylamide SDS-PAGE and electro-transferred on nitrocellulose membrane (GE Healthcare Whatman). The membranes were subsequently probed with anti-ABCH2 antibody at 1:250 dilution or anti-β-tubulin (Cell Signaling) at 1:500 dilution in a 1% skimmed milk/TBS-0.1%Tween buffer. Antibody binding was detected using goat anti-rabbit or anti-mouse IgG coupled to horseradish peroxidase (Cell

Signaling) (dilution: 1:5000 in 1% skimmed milk in TBS-Tween). Visualization was performed using a horseradish peroxidase sensitive ECL western blotting detection kit (SuperSignal West Pico PLUS Chemiluminescent Substrate, ThermoScientific) and the result was recorded using Chemidoc Imaging System (Bio-Rad Laboratories).

### Tissue dissection and relative expression estimation

To determine *ABCH2* expression levels in different tissues, 3–5 day-old female *An. coluzzii* were dissected, RNA-extracted and subjected to RT-qPCR analysis. Heads, thoraces, guts with the attached Malpighian tubules and ovaries, abdominal walls and legs were separated into TRIzol Reagent. Each sample contained 20 dissected tissues in 200 μl reagent, and three replicas per tissue were prepared according to manufacturer's instructions. After elution, RNAs were subjected to DNase I treatment (ThermoFisher Scientific, DNase I, RNase-free). The quantity of DNase-treated RNAs was calculated using a Nanodrop spectrophotometer and equal quantities (1μg) from each sample were used for cDNA synthesis. This was carried out using EnzyQuest Reverse Transcriptase (Minotech Biotechnology) and oligodTs according to the guidelines and the reaction products were used for RT-qPCR. RT-qPCR was performed in a BIO-RAD cycler in the following conditions: 3 min at 95˚C, 40 cycles of 15 seconds at 95˚C and 30 seconds at 60˚C and the analysis included three biological and two technical replicates within each biological replicate. Relative expression was normalized with *An. coluzzii Ribosomal Protein S7* (*S7*) housekeeping gene. The relative expression of *ABCH2* among different tissues was measured against abdominal wall samples. Graphs were produced and statistically analyzed with GraphPad Prism software version 8 using Student's t-test.

### Cryosectioning, immunofluorescence and confocal microscopy

Localization analysis of ABCH2, was conducted in longitudinal leg and head appendage cryosections using a specific antibody raised against ABCH2 and E-cadherin antibody, as an epithelial cell marker. Legs and head appendages from 3–5 day-old *An. coluzzii* were dissected and placed for 2 hours in PCR tubes containing 4% PFA in 1x PBS buffer for fixation. After removal of fixative, legs were incubated overnight with 30% sucrose/PBS buffer at 4˚C. Legs were then immobilized in eppendorf lids covered with Optimal Cutting Temperature compound (O.C.T—Tissue-Tek SAKURA) and placed at -80˚C. 5 μm longitudinal leg sections were obtained in cryostat (Leica CM1850UV) and were placed on superfrost microscope slides (Thermo Scientific). The slides were washed (3 x 5 min) with 0.02% Tween/PBS, followed by a 10 minute incubation with 0.03% Triton/PBS. After 1 hour blocking with 1% Fetal Bovine Serum in 0.03% Triton/PBS, the slides were incubated overnight with the ABCH2 antibody in 1:500 dilution, at 4˚C. The next day goat anti-rabbit or goat anti-mouse (Alexa Fluor 488, Molecular Probes) were used in 1:1000 dilutions. TO-PRO-3 dye (Molecular Probes) was used for nuclei staining after RNAse A (Invitrogen Ambion) treatment in a 1:1000 dilution for 5 min. Observation and image attainment were carried out at a Leica SP8 laser-scanning microscope, using a 40x- magnification lense.

### dsRNA design, generation, nano-injections and silencing efficiency

For gene silencing, a dsRNA construct targeting ABCH2 was designed, synthesized and injected into newly-emerged female adult *An. coluzzii*. The silencing efficiency was estimated 72 hours after injection in legs and head appendages using RT-qPCR and western blot analysis. Primer sequence (S1 Table) for dsRNA synthesis of *ABCH2* were designed using PrimerBlast and they amplify a product of 525 bp (primer sequence, S1 Table). The T7 promoter sequence (5' TAATACGACTCACTATAGGG 3') was added at the 5' end of both forward and reverse

oligos. VK7 cDNA was used as template for PCR amplification using Phusion High-Fidelity DNA Polymerase (New England Biolabs), following manufacturer's instructions. Specific amplification was verified on a 1.5% agarose gel and the rest of the reaction was purified using Macherey-Nagel Nucleospin Gel and PCR Clean-up Kit. The purified product was used as template for dsRNA synthesis using HiScribe T7 High Yield RNA Synthesis Kit (New England Biolabs) with subsequent purification with MEGAclear Transcription Clean-Up Kit (Ambion). The purified dsRNAs were diluted to a 3 μg/μl concentration and 69 nl were injected into $CO_2$-anaesthetized female 0-day mosquitoes. The intrathoaracic dsRNA injections were performed using a Nanoinject II Auto-Nanoliter Injector (Drummond Scientific Company). As control a 500bp dsRNA prepared form the non-endogenous *green fluorescent protein* (*GFP*) gene was used after similar preparation from a GFP plasmid template (primer sequence, S1 Table). Injected mosquitoes were placed in cups and kept in insectary conditions with 10% sugar impregnated cotton wool for 72 hours. After that their legs were dissected, RNA extracted, cDNA synthesized and RT-qPCR was conducted, as described in detail above. Primers for qPCR were designed out of the region-targeted by dsRNA resulting in 150–200 bp product (primer sequence, S1 Table) and according to standard curve construction their efficiency was 96%. Silencing efficiency for each dsRNA was estimated after comparison of relative expression of each gene of interest in dsRNA-injected against relative expression levels in ds*GFP*-injected female mosquitoes. The ds*ABCH2* resulted in about 80% reduction of *ABCH2* transcript level. Potential non-specific targeting of the dsABCH2 construct against the other two ABCH transporters, ABCH1 and ABCH3 was also tested (S3 Fig). Primer sequences (S1 Table) were designed out of the aligned region with the dsABCH2 construct to avoid any potential amplification of the provided dsRNA. Graphs were produced and statistically analyzed using GraphPad Prism software version 8 using Student's t-test. For protein estimation, polypeptide extraction from legs/appendages and western blot analysis were performed as described in section "Western blot analysis".

## Deltamethrin toxicity assays

Deltamethrin toxicity assays were carried out on ds*GFP* and ds*ABCH2*-injected female mosquitoes, in WHO tubes. 0-Day females were injected with ds*GFP* or ds*ABCH2* and then kept in cups for 72 hours as described above. After this, they were exposed to 0.016% deltamethrin (the estimated LC50 for this VK7 strain (VK7-LR) [7], using insecticide impregnated papers in WHO tubes. The exposure lasted for 1 hour and after this the number of knocked-down mosquitoes were recorded. This was followed by 24 hour recovery in control tubes in insectary conditions. A mosquito was classified as dead or knocked down if it was immobile, unable to stand or take off. Specifically, four independent injection-exposure experiments with different mosquito batches were carried out. In every replicate ds*GFP*- and ds*ABCH2* mosquitoes were injected concomitantly, to ensure comparable injection, insectary and bioassay conditions. The number of mosquitoes used in each replicate and each condition, and the numbers and percentages of knocked-down and dead mosquitoes are shown in the S4A and S4B Fig. Graphs were produced and statistically analyzed using GraphPad Prism software version 8 and statistical analysis was performed with two-sided Fisher's exact test (S4C Fig).

## [14]C penetration in ds*ABCH2* and ds*GFP* mosquitoes

To estimate penetration of the insecticide, the internal and external deltamethrin in ds*GFP*- and ds*ABCH2*-injected mosquito pools were counted after [14]C-Deltamethrin exposure. The external deltamethrin, removed with hexane rinsing and the internal deltamethrin of homogenized mosquitoes were used for estimation of [14]C penetration in the two experimental

conditions. Specifically, penetration was assessed as previously described in [45]. Briefly newly-emerged ds*ABCH2*- and ds*GFP*-injected mosquitoes, three days post injection, were subjected to 4 minute contact on 0.01% [14]C-deltamethrin paper, prepared using standard WHO bioassay protocols (http://apps.who.int/iris/bitstream/handle/10665/64879/WHO_CDS_CPC_MAL_98.12.pdf;jsessionid=5BF98105B548CDE77F152EE4713EE949?sequence=1). Mosquitoes were collected in glass vials and three 1 minute-hexane washes were performed. After that they were homogenized in PBS. In all samples 10ml of liquid Scintillation Counting Mixture (Ultima Gold;6013326; PerkinElmer) were added and they were measured on a beta counter (LS1701; Beckman). Internal counts per minute correspond to the PBS-homogenized mosquitoes, while external counts per minute correspond to the average of the counts per minute of the three hexane washes. Penetration was calculated as the ratio of the internal to the total counts per minute in the two conditions. Three biological replicates were performed with 116 mosquitoes tested in total in each condition (S2 Table).

## Phylogenetic analysis

Multiple sequence alignment was performed using Mafft v7.310 [66] with default parameters. Alignments were trimmed using trimAl [67] and converted to a phylip format file using a custom Bash script. The phylogenetic tree was built under the maximum likelihood optimality criterion using IQ-TREE2 [68] with the following parameters "*-alrt 5000 -bb 5000 -m MFP*". Tree visualization was performed using Evolview3 [69].

## Extraction of cuticular lipids, cuticular hydrocarbons (CHCs) fractionation, identification and quantitation

For CHC analysis, dried legs of injected ds*ABCH2* and ds*GFP* mosquitoes were extracted, identified by gas chromatography-mass spectrometry (GC-MS) and quantified by GC-flame ionization detection (FID). 3 days after dsRNA nano-injections performed in newly emerged females, 95 ds*ABCH2* and 103 ds*GFP* female mosquitoes (3 replicates/ 30–35 mosquitoes per replicate) were dried at Room Temperature for about 48 h. Legs of dry mosquitoes were separated from the rest of the body. Legs of dried mosquitoes were pooled and weighed using a balance scale. Each replicate was approximately 1 μg and the number of mosquitoes used was recorded to allow normalization per mosquito. CHC analysis was carried out in VITAS-Analytical Services (Oslo, Norway) as described in [70]. Silencing for this batch of injections was also tested with western blot analysis (S7 Fig). Statistics were analyzed using GraphPad Prism software, version 8 and statistical analysis carried out with Student's t-test.

## ABCH2-overexpressing Sf9 insect cells, membrane preparations and ATPase activity estimation

The recombinant expression of ABCH2 was achieved in Sf9 cells using the Baculovirus expression system. Isolated membrane vesicles of ABCH2- and control-expressing cells were used in a colorimetric ATPase assay, in which the basal transporter activity and the response upon deltamethrin were calculated. More specifically, ABCH2 was expressed in *Spodoptera frugiperda Sf9* insect cells using the Pfastbac1 vector, which was synthesized *de novo* (GenScript) and *ABCH2* ORF was subcloned in between BamHI and EcoRI restriction enzyme sites. The sequence was codon optimized for *S. frugiperda* using GenSmart codon optimization tool (GenScript). Recombinant baculoviruses encoding *ABCH2* cDNA were generated with BAC-TO-BAC Baculovirus Expression Systems (Invitrogen), following manufacturer's instructions. 2μg of Bacmid DNA, mixed with Escort IV Transfection Reagent (Merck) were used to

transfect $5 \times 10^5$ Sf9 cells in 1ml of SF900 II SFM growth medium (ThermoFisher scientific). The complex was incubated for 45 min/RT and the lipid complexes were added dropwise to the appropriate well and were incubated at 27°C for 6 hours. After that, another 1ml of medium, supplemented with antibiotics (2x penicillin and streptomycin) and 20% FBS (ThermoFisher scientific) were added. After 24 hours DNA:lipid complexes were removed and 2ml of supplemented growth medium were added to the cells which were subsequently incubated at 27°C for 72 hours. Medium, containing the virus stock were removed and stored, while cells were also collected by pipetting in ice-cold 1 X PBS. Cell pellets (after 3000g/5min/4°C centrifugation) were resuspended in RIPA supplemented with protease inhibitors followed by a 10,000g/10 min /4°C centrifugation step. The supernatant was prepared for western blot analysis with the addition of 5x Sample Buffer (SB).

After validation of recombinant protein expression of expected size in Sf9 cells and determination of viral stock titers using baculoQUANT ALL-IN-ONE (GenWay), according to manufacturer's instructions, infection was performed. For infection $10^6$ cells/ml were seeded in T75 Flasks in a final volume of 15 ml antibiotic and FBS supplemented growth medium. After 4–5 hour incubation at 27°C, viruses were added at the desired MOI and infected cells were incubated for other 96 hours. Along with the ABCH2 infections, cells infected with an empty Bacmid and an unrelated membrane protein served as negative controls. This unrelated protein is a cytochrome P450 protein (*An. coluzzii* CYP4G16), which is anchored to the membranes with a single N-terminal transmembrane domain [70] and has no known transport or ATPase activity.

After incubation the cells were harvested and microsomes prepared. Following centrifugation at 2,000g/3min/4°C, cell pellets were homogenized using a glass-Teflon tissue homogenizator in a mannitol-containing buffer as described in [71]. Final ultracentrifugation step at 100,000g/1hour/4°C (Beckman AirFuge CLS Ultracentrifuge) resulted in pellets which were resuspended in the same buffer [71]. After estimation of total protein content of the membrane preparations by Bradford protein assay using BSA to generate a linear control standard curve (0.5–20μg) the membranes were further diluted to 1–2 μg/μl to be used for downstream experiments or stored at -80°C as 20μl aliquots.

The ATPase activity of the Sf9 membranes was estimated with a colorimetric assay by measuring inorganic Pi released from ATP hydrolysis as described by [71,72], with some modifications. Briefly, we prepared a malachite green solution (340 mg of Malachite Green (Sigma) in 75 ml deionized water) and an ammonium molybdate solution (10.5 g ammonium molybdate in 250 ml 4N HCl). Mixing the two solutions followed by filtration through paper resulted in the malachite green stock solution that was stored at 4°C. The malachite working solution (MGWS) was freshly prepared by adding 20% Triton-X-100 in malachite green stock solution to a 0.1% final concentration. Phosphorous standard solutions incubated with MGWS (50 μl of each with 0.8 μl MGWS) were used to generate linear control standard curves (0–20 nmoles Pi) [73]. Pi standards were incubated in MGWS for 5 minutes and then 100ul 37% citric acid was added to stop the reaction.

The reactions were then incubated for 40 minutes at Room Temperature and absorbance was measured at 630nm. For estimation of Pi liberation in membrane preparations, reactions were set up using 0.5 μl of 0.1 M ATP, 2 μl of 10mg/ml BSA, 10 μg of membrane protein, 5 μl of a 10X buffer (500mM Tris/HCL Ph = 8, 50mM MgCl2, 50mM KCl and 10mM DTT) and deionized water was added up to 50ul. The reactions were incubated in a water-bath at 37°C for 15 minutes to allow ATP hydrolysis to occur and afterwards they were mixed with 0.8μl freshly prepared MGWS. Then a further 5 minute incubation was performed and terminated by adding 100 μl of 37% citric acid to avoid further ATP hydrolysis. The reactions were kept at RT for 40 minutes and then absorbance measured at 630nm.

In every replicate empty, ABCH2 or unrelated protein membranes were tested simultaneously. The empty Bacmid was used to remove the Sf9 non-specific ATPase background. In every ATPase experiment two control solutions were used: a reaction containing the buffer without proteins and two reactions containing no ATP, one for each membrane preparation. We used the first control as blank which was subtracted at every absorbance value and thereafter the liberated phosphate was quantified based on the Pi standard curve. As this comparison is based on total protein content estimation, whole membrane protein extracts were tested on SDS-Page and the separated polypeptides were visualized with Coomassie Blue (S9C Fig). According to the staining ABCH2 was weakly expressed in Sf9 membranes since an induced band was not apparent when expressing and control membranes were compared.

For ATPase experiments with deltamethrin, the same set-up was used. Deltamethrin was resuspended in DMSO/ethanol (1:1 v/v). A working stock of 0.6 mg/ ml was used and 5μl deltamethrin was added in each reaction (3μg-120μM). Each reaction contained 10μg of total protein from empty and ABCH2 membrane isolations. Reactions containing only DMSO/ethanol were also processed. The empty/DMSO/ethanol and empty/deltamethrin Pi/μgr of total protein/minute of reaction estimated values were subtracted from the ABCH2/DMSO/ethanol and ABCH2/deltamethrin accordingly, thus removing the background and allowing estimation of ABCH2-deltamethrin stimulation over the deltamethrin controls (S5 Table).

## Estimation of ABCH2 content in Sf9 membranes

For the approximate estimation of ABCH2 content in Sf9 membrane preparations, we used a His-tagged ABCH2 construct and a purified His-tagged protein in western blot analysis. In detail, parallel to the infection with ABCH2 bacmid, infection with a His-tagged ABCH2 bacmid was carried out and membranes were prepared to allow quantification based on a known quantity of his-tagged purified protein. Total protein content of His-ABCH2 expressing membranes was quantified using Bradford protein assay and known quantities ran on the same gel with a known concentration of either a His-tagged, purified, unrelated protein or the ABCH2 membranes. Upon separation, the polypeptides were electro-transferred on two separate nitrocellulose membranes, which were blotted with anti-His or anti-ABCH2 respectively (S11 Fig). Then, imageJ was used for pairwise band density analysis (His-ABCH2 and His-protein, His-ABCH2 and ABCH2). Based on the estimation described in S3 Table, ABCH2 was about 11 ng per μg of total protein, which corresponds to 0.12 pmoles (based on the expected molecular weight of the transporter, 85kDa).

## Structural analysis and molecular docking simulations

The amino acid sequence of ABCH2 (Uniprot:A0A6E8VHH0) was submitted to the SWISS--MODEL in the automated protein modelling server provided by the GlaxoSmithKline center, using its standard settings [74]. The same software was also used to identify the best template for the homology-based modelling [74]. The human homodimeric ABCG1 transporter involved in cholesterol trafficking [46] was identified as the closest homologue. For both the ABCG1 structure and the ABCH2 predicted model, the Protein Interaction Calculator (PIC) webserver was used to predict intermolecular interactions [47]. All Molecular Docking studies were performed using Autodock Smina, a fork of Autodock Vina software (includes changes from the standard Vina version 1.1.2) [50], and visualized using the PyMOL Molecular Graphics System (Version 2.0.6 Schrödinger, LLC). The structure used as receptor for the ABCG1 transporter was obtained from Protein Data Bank (PDB code: 7R8D), [75], and the 3D structures of the ligands were obtained from PubChem database [76]. The nature of all protein-ligand interactions was identified using Protein-Ligand Interaction Profiler (PLIP) [76].

## Supporting information

**S1 Fig. *ABCH2* relative expression in non-induced legs, 1hour-deltamethrin induced legs and bodies.** Mean + SEM of three biological replicates per condition (n = 45 per condition). *ABCH2* expression is 2.8-fold higher in induced legs, compared to non-induced (P value = 0.0487, *) and 8.6-folds higher in induced legs compared to induced bodies (P value = 0.0192, *). Bodies represent the whole female mosquitoes, lacking the legs.
(TIF)

**S2 Fig. ABCH2 expression in different tissues.** A. Relative expression levels of *ABCH2* in different dissected tissues normalized against abdominal walls of 3–5 Day old female VK7 mosquitoes. Bar graphs represent mean values; error bars standard mean error; n = 3 biological replicates and B. Western Blot using specific antibody against ABCH2, depicting ABCH2 in different dissected tissues of 3–5 day old female VK7. Each band corresponds to the tissue of the bar above. C. Western blot analysis indicating ABCH2 presence in head is exclusively attributed to sensory appendages. β-tubulin was used as loading control.
(TIF)

**S3 Fig. Relative expression of *ABCH*s in the ds*GFP* and ds*ABCH2*-injected mosquitoes.** For each gene, expression is normalized against ds*GFP*. ABCH2 relative expression in ds*ABCH2*-injected mosquitoes is reduced by 81.9% (*P*-value = 0.0085, **), ABCH3 is reduced by 41.4% (*P*-value = 0.134, ns) and ABCH1 expression is reduced by 29.7% (*P*-value = 0.354, ns).
(TIF)

**S4 Fig. Number of knocked-down (KD), alive and dead mosquitoes used in deltamethrin toxicity assay.** A. Numbers of ds*ABCH2* and ds*GFP* mosquitoes exposed in deltamethrin per biological replicate, B. Graphical depiction of total ds*ABCH2* and ds*GFP* alive-KD (left panel) and alive-dead (right panel) after 1 hour and 24 hours post deltamethrin exposure respectively, C. Statistical analysis using two-sided exact Fisher's test: *P*-value and effect size (odds ratios and reciprocal odds ratio) values are shown for KD and mortality after deltamethrin exposure.
(TIF)

**S5 Fig. ABCH2 subcellular localization is predicted to be on the plasma membrane using DeepLoc-1.0: Eukaryotic protein subcellular localization predictor.**
(TIF)

**S6 Fig. ACON003680 is one-to-one ortholog of *snustorr*.** Phylogenetic analysis of ABCH transporters from *Anopheles coluzzii* (ACON), *Anopheles gambiae* (Ag), *Drosophila melanogaster* (Dm), *Plutella xylostella* (Px), *Nezara viridula* (Nv), *Tribolium castaneum* (Tc) and *Locusta migratoria* (Lm). Tree was created under the LG+R5 substitution model with 5,000 bootstraps and was rooted using the *Danio rerio* ABCH gene as an outgroup. Nodes with bootstrap support < 50% and between 50% and 75% are indicated with light grey and grey circles respectively. Nodes with bootstrap support greater than 75% are indicated with black circles.
(TIF)

**S7 Fig. Western blot analysis of a portion of the injected mosquito batch that was used for CHC analysis.** Legs from 10 ds*GFP* and 10 ds*ABCH2* were used to validate the silencing.
(TIF)

**S8 Fig. Relative abundance of cuticular hydrocarbons (CHCs) identified in legs of dsRNA injected mosquitoes (dsABCH2 and dsGFP-control).** Relative abundances in % area are

depicted for each one of the identified CHCs. Mean of 3 biological replicates +SEM.
(TIF)

**S9 Fig. *In vitro* expression of ABCH2 in Sf9 membranes.** A. Western blot analysis using ABCH2 antibody in whole cell extracts of ABCH2- and Empty Bacmid-infected Sf9 cells, B. Western blot analysis using ABCH2 antibody in cytoplasmic fractions (supernatant). In all cases B-TUBULIN was used as loading control. C. Coomassie blue staining in ABCH2 and Empty membrane fractions.
(TIF)

**S10 Fig. Functional expression in Sf9 IMVs.** ATPase activity of membrane preparations based on the malachite green colorimetric assay. A. Mean + SEM of six biological replicates in ABCH2 and empty-bacmid expressing membranes. Mean + SEM; $Mean_{(EMPTY)} = 32.04+ 4$ and $Mean_{(ABCH2)} = 52.4+ 3.32$ for n = 6 biological replicates; *P*-value = 0.0021 (**). B. 3 biological replicates in unrelated protein-expressing membranes and ABCH2 membranes. The average of 2–3 technical replicates for each biological is presented in bars. Mean + SEM; $Mean_{(UNRELATED\ PROTEIN)} = 33.74+ 5.67$ and $Mean_{(ABCH2)} = 57.94+ 3.68$; *P*-value = 0.0238 (*). Empty-bacmid values were used to subtract the Sf9 cell background (Fig 4D).
(TIF)

**S11 Fig. Western blot analysis for the estimation of the amount of ABCH2 in the ATP hydrolysis reactions.** Sf9 infections using ABCH2 and His-ABCH2 bacmids were carried out simultaneously. After isolation of membrane fractions and total protein content estimation using Bradford, the indicated calculated amounts were analyzed by SDS-PAGE and blotted using an ABCH2 specific antibody. At the same time the His-ABCH2 was analyzed simultaneously together with a purified His-tagged control protein to allow estimation of the His-ABCH2 amount using the anti-His antibody. According to densitometry analysis using ImageJ and subsequent calculation (Supplementary file 3), we estimate ABCH2 to be about 11 ng per μgr of total membrane protein, which corresponds to 0.12 pmoles of ABCH2 per μgr of total protein.
(TIF)

**S1 Table. ABCH2 primers for dsRNA construction and RT-Qpcr.**
(XLSX)

**S2 Table.** A. [14]C measurements of three biological experiments are presented: External, internal, total and % penetration of [14]C-deltamethrin after 4 minutes of contact in ds*GFP* and ds*ABCH2* mosquitoes. Penetration corresponds to the percentage of internal counts compared to total counts at this time point. B. Statistical analysis was performed with paired t-test (*) and statistical details of the comparison (P value = 0.0037, **).
(XLSX)

**S3 Table. Calculation details for the estimation of ABCH2 quantity (ng) used in ATPase assay.** Known quantities of a His-tagged purified protein were blotted with His-ABCH2 and ABCH2 membrane preparations (S11 Fig). ImageJ was used for pairwise band density analysis. The density values and the μgr of total protein were used for standard curve construction. The standard curves were subsequently used for estimation of the ng of ABCH2 per μgr of total protein.
(XLSX)

**S4 Table. The estimated ABCH2 pmoles in our preparation (S7 Fig) allows us to extract the pmoles of Pi per pmole of ABCH2 per minute of reaction.** As shown in red, the

hydrolysis rate is ~178 pmoles of Pi/ pmole of ABCH2/ minute of reaction. Even if our estimation is indirect and prone to errors, the rate value is substantial indicating that ABCH2 undergoes consecutive ATP hydrolysis rounds.
(XLSX)

**S5 Table. ATPase assay using deltamethrin diluted in DMSO in ABCH2 and empty-bacmid expressing membrane preparations.** 120μM deltamethrin was added in 50ul reaction containing 10μgr membranes (approximately 100 ngr ABCH2). The values of pmoles Pi/ μgr of total protein/ minute of reaction and the vales of pmoles Pi/ pmoles of ABCH2/ minute of reaction from two biological replicates (3 technical replicates each), are shown. The addition of deltamethrin results in ~1.6 folds of stimulation of the basal activity.
(XLSX)

**S1 File. The inter-protein interactions of the solved or modelled structures, retrieved form the Protein Interaction Calculator (PIC) webserver (47).** The standard cut-off distances for all interaction types (hydrophobic, hydrogen-bonds, ionic, aromatic, cation-Pi) were used. The interactions were also manually inspected using Pymol (Molecular Graphics System, version 1.6 Schrödinger, LLC).
(PDF)

## Acknowledgments

We thank Dr Inga Siden-Kiamos (IMBB/FORTH) and Lefteris Spanos (IMBB/FORTH) for providing E-cadherin antibody. We would also like to acknowledge Dr Linda Grigoraki (IMBB/FORTH) for critical reading of the manuscript.

## Author Contributions

**Conceptualization:** Mary Kefi, Vasileia Balabanidou, John Vontas.

**Formal analysis:** Mary Kefi.

**Funding acquisition:** Mary Kefi.

**Investigation:** Mary Kefi, Chara Sarafoglou, Jason Charamis, Giorgos Gouridis.

**Methodology:** Mary Kefi, Vasileia Balabanidou, Chara Sarafoglou, Jason Charamis, Giorgos Gouridis.

**Project administration:** Mary Kefi, John Vontas.

**Resources:** Hilary Ranson, John Vontas.

**Supervision:** John Vontas.

**Validation:** Vasileia Balabanidou, Giorgos Gouridis, John Vontas.

**Visualization:** Mary Kefi.

**Writing – original draft:** Mary Kefi.

**Writing – review & editing:** Vasileia Balabanidou, Chara Sarafoglou, Jason Charamis, Gareth Lycett, Hilary Ranson, Giorgos Gouridis, John Vontas.

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
