## [Decision Letter · Decision Letter 0]

18 Apr 2023

Dear Dr Kefi,

Thank you very much for submitting your manuscript "ABCH2 transporter in the first line of defense protects malaria vectors from pyrethroids" for consideration at PLOS Pathogens. As with all papers reviewed by the journal, your manuscript was reviewed by members of the editorial board and by several independent reviewers. In light of the reviews (below this email), we would like to invite the resubmission of a significantly-revised version that takes into account the reviewers' comments.

We cannot make any decision about publication until we have seen the revised manuscript and your response to the reviewers' comments. Your revised manuscript is also likely to be sent to reviewers for further evaluation.

Sincerely,

Kenneth D Vernick

Academic Editor

PLOS Pathogens

Margaret Phillips

Section Editor

PLOS Pathogens

Kasturi Haldar

Editor-in-Chief

PLOS Pathogens

orcid.org/0000-0001-5065-158X

Michael Malim

Editor-in-Chief

PLOS Pathogens

orcid.org/0000-0002-7699-2064

Reviewer's Responses to Questions

**Part I - Summary**

Reviewer #1: This MS describes the functional study of an ABC-transporter (ABCH2) from the malaria vector Anopheles coluzzi. In particular, the potency of this ABC-transporter to mediate the toxicity of deltamethrin (a pyrethroid insecticide widely used on bednets) is investigated through a series of descriptive and functional experiments.

First, using RT-qPCR and western blotting, the authors showed that ABCH2 is strongly expressed in mosquito legs and head appendices as compared to other tissues, in line with its potency to interact with the insecticide following an epidermal/tarsal contact. Then, in vivo RNAi silencing (intrathoracic injection of ABCH2 dsRNA into adult females) was used to show that ABCH2 expression is associated with an increased deltamethrin survival and a decreased C14-deltamethrin penetration. Immunofluorescence was then used to show that ABCH2 protein is mainly localized in the sub-cuticular epithelia of legs and antennae. Comparing total cuticular hydrocarbon (CHCs) content and CHCs composition between dsABCH2- and dsGFP-injected mosquitoes did not support the hypothesis of ABCH2 being involved in CHC deposition. In silico modelling and docking simulations supported ABCH2 acting as a homo-dimeric active transmembrane deltamethrin transporter. In vitro experiments using recombinant ABCH2 showed an increased ATPase activity in presence of deltamethrin supporting its ability to actively transport the insecticide. Altogether, these experiments support the role of this ABC-transporter in mediating deltamethrin toxicity in the malaria vector An. coluzzi.

Overall, this MS presents a huge amount of work and provides interesting functional data about the role of ABC transporters in mediating insecticide toxicity in malaria vectors. The MS is well written, figures are well designed and supplementary materials are overall informative.

However, I found the whole story slightly over-sold with some experimental data possibly over-interpreted (see specific comments below). This feeling is also supported by a very straight forward discussion which, in my view, does not always mention the technical biases potentially affecting some of the experimental data presented and their interpretation. The discussion would also benefit from an additional section replacing the findings into the broader context of the response/adaptation of mosquitoes to insecticides. For instance, discussing the relative importance of this mechanism versus other known insecticide resistance mechanisms and its adaptive value (as opposed to a non-adaptive effect in mediating insecticide toxicity) would be of interest for the community.

Reviewer #2: While malaria vector control still relies heavily on the use of insecticides deployed as long-lasting insecticidal nets and indoor residual spraying, insecticide resistance increasingly threatens the sustainability of current mosquito control programmes. Understanding the mechanisms conferring insecticide resistance in the malaria vector may help to find new targets in the development of alternative active ingredients. One of the mechanisms of insecticide resistance is resistance due to reduction in insecticide uptake through the insect’s cuticle, which is the topic of the present article. Specifically, the study investigated the role of the ATP-Binding-Casette (ABC) transporter ABCH2 in conferring resistance to the pyrethroid insecticide deltamethrin in the sub-Saharan malaria vector, Anopheles coluzzii. The methods used are a combination of many complementary approaches, including RNAi-mediated silencing of ABCH2 in a pyrethroid-resistant A. coluzzii mosquitoes from a laboratory colony, standard insecticide susceptibility assays to characterise the phenotype of silenced and control groups, measurements of 14C-labelled deltamethrin between silenced and control groups, immunolocalisation of ABCH2 in legs and head appendages with immunostaining and confocal microscopy, measurement of cuticular hydrocarbons contents between the groups, determination of gene expression levels using RT-qPCR, use of in silico protein models as well as ATPase activity measurements in heterologously expressed ABCH2. The overall emerging picture from the results is that deltamethrin is a substrate of ABCH2 that then actively export deltamethrin through hydrolysis of ATP.

A strength of the study is its holistic approach to understand the mechanism of ABCH2 in conferring resistance to deltamethrin in A. coluzzii. Where I found the study is less convincing is to what degree ABCH2 plays a role in shaping the resistance phenotype. While the main conclusion of the study is that ABCH2 is a ‘key’ regulator of deltamethrin toxicity, I feel the data do not show to what extent ABCH2 indeed plays a role in deltamethrin detoxification since effect sizes are not provided and some of the experiments lack sufficient replication (see comments below). Therefore, it may not be concluded that ABCH2 is a ‘key regulator of deltamethrin toxicity’. Nevertheless, the study highlights an intriguing aspect of insecticide detoxification in mosquitoes and opens the door for further studies and, as such, it is novel and merits sharing with the wider research community.

**Part II – Major Issues: Key Experiments Required for Acceptance**

Reviewer #1: Considering that several experiments rely on the differential mortality observed between dsABCH2- and dsGFP-injected mosquitoes, I suggest strengthening the presentation of these data. For instance, it is mentioned that there are 3 ABCH transporters in An. coluzzi (with high sequence homology) while the specificity of dsABCH2 against other ABCH transporters is not presented. Considering the RT-qPCR data presented, I believe that adding this does not represent a major challenge as the same cDNA samples as those used for ABCH2 expression data can be used. From the method section, I understand that total RNA was extracted from legs so it might not be possible to assess the effect of dsABCH2-injection on all three ABCH transcripts in various tissues, though at least presenting such data for legs will be informative. This is a major point because providing such data will eliminate any doubt regarding the specificity of the phenotype observed upon dsABCH2 injection and strengthen subsequent experiments and general conclusions.

In addition, although the mortality difference between dsGFP- and dsABCH2-injected mosquitoes appears significant, the underlying raw data should be made available to the reader (as supplementary information) in order to support the statistical approach presented. Indeed, Fig 1 does not show replicate data points (only means and total number of mosquitoes), and Fig1 legend does not precise if SEM or SD are shown. The way experiment was replicated should also be better described in the method section (lines 478-481): It is mentioned ‘14-22 individuals x 4 replicates per dsRNA’, but not sure what is considered as a replicate (same injected-mosquito batch, or totally independent injection experiments with a different mosquito batch?) and how many mosquitoes per tube were used per replicate. Again, providing raw data as supplementary information (number mosquitoes KD, dead, alive for each test tube and each replicate) will strengthen the study.

In the same line but of lesser importance, the experiment looking at CHCs deposition also involves comparing dsABCH2- and dsGFP-injected mosquitoes. As this experiment does not show any significant difference between the two conditions (thus allowing to reject the hypothesis of ABCH2 being involved in CHC deposition), one would expect these CHC data to be backed up with RT-qPCR data showing a significant knock down of ABCH2 expression on the same injected mosquito batch. If not technically possible, this should be mentioned in the results and/or the discussion as this may affect data interpretation.

Reviewer #2: A key experiment, which I find is very important to substantiate ABCH2’s role in deltamethrin detoxification, is the measurement of the change in deltamethrin content between ABCH2 silenced and the GFP control mosquitoes using 14C-deltamethrin. It is unfortunate that data from only two replicates per group are available which makes it impossible to estimate the effect size of silencing ABCH2 (i.e. the difference reported is statistically not significant). As this is a key experiment to make a strong case for the function and role of ABCH2, the manuscript would substantially improve by repeating this assay with more replicates and estimating the effect size.

Similar experimental design issues are present in other parts of the study. Although the difference between CHC of legs from dsABCH2 and dsGFP may be small, making a claim that CHS is not affected by ABCH2 silencing based on a sample size of 3 is very bold. In this case, I would argue that a lack of evidence is not an evidence of lack as the experiment is hugely underpowered. Therefore, either more replicates are needed to make such a statement - together with a power calculation to indicate what difference may be detected - or the authors ought to be more cautious in the interpretation of these results.

Finally, I would like to see a critical discussion of alternative hypothesis and weaknesses of the study in the Discussion section. For instance, is it feasible that ABCH2 plays an important role (other than detoxification of deltamethrin) that would lead to higher mortality when silenced? Where do the authors feel the study has weaknesses in terms of the methods applied?

**Part III – Minor Issues: Editorial and Data Presentation Modifications**

Reviewer #1: - Title: The title seems a bit overselling as the MS does not show any evidence that the mechanism studied differentiates resistant from susceptible populations. Consider changing to something more realistic such as ‘ABCH2 transporter mediates deltamethrin uptake and toxicity in the malaria vector An. coluzzi’.

- line 54-55: ref (8) repeated twice

- line 73: TBD -> TMD?

- line 82: Maybe I am mistaken, but I think that ref (6) and (10) do not support the over-transcription of ABCH2 in insecticide resistant populations nor its induction after insecticide exposure. The following sentence was taken from ref (10): ‘The down-regulated transcripts include a number of cuticular proteins and the ABC transporter ABCH2, a half transporter whose role in insects is poorly characterised [18]’. This looks as overselling the rational of the study.

- line 87: I did not find mention of ABCH expression in ref (4). In addition, in ref (6) the An. coluzzi ortholog of An. gambiae ABCH2 (see the phylogenetic analysis in supplementary material) was not overexpressed in legs (as opposed to ABCH3), neither induced by deltamethrin (as opposed to ABCG11, ABCE2, ABCC2 and BCG2) or associated with constitutive deltamethrin resistance (no ABC transporter mentioned). Again, this looks as overselling the rational of the study. In this regard, a screening of the recent literature in An. gambiae/coluzzi for a synthesis of ABCH gene expression data in regards of insecticide resistance and xenobiotic exposure may be of interest as this may consolidate the rational of the study but also feed the discussion about the role of mosquito ABCH transporters in their response to insecticides.

- lines 95-96: same comment as above (line 87).

- Fig4D. Why not mentioning which protein was used as control?

- line 392: Explain the rational of using the VK7-LR colony (instead of the VK7 colony). You may also want to describe the remaining resistance mechanisms identified in this colony based on previous work (kdr mutation frequency, other, …).

Reviewer #2: Generally, the manuscript has many typos. I recommend careful revision of the text by the native Anglophone co-authors.

Spell out the scientific names on first mentioning. For example, line 50 (‘Anopheles coluzzii’), line 59 (‘Bemisia tabaci’ and ‘Plutella xylostella’).

Lines 116 – 120 and Figure 1C and D. Using Student’s t-test to compare mortality rates is unsuitable. Use a binomial test, instead. It would be useful to know the odds ratios incl. 95% confidence intervals to have an idea of the effect size.

Figure 1A: This figure seems a bit odd to me. I would have expected either only one bar chart with the fold changes in ABCH2 compared to dsGFP or two bars with the actual expression levels. The number of mosquitoes in the brackets is not readable. I would also suggest showing the 95% confidence limits rather than SEM as it is more intuitive.

Figure 1C and D: see comments with regards to the statistical analysis above. I am not too familiar with RNAi experiments. Is it usual that we expect such high mortality rates in the controls? Or is the laboratory colony a rather susceptible one (i.e. the insecticide dose chosen too high)? Please provide an explanation in the text / discussion.

Figure 1E: The supposed difference between dsGFP and dsABCH2 is inflated because the y-axis does not start at 0%. Adjust the y-axis with 0% as the lower limit.

Figure 2: Increase the size of the labels on the images. The scale bars, for example, cannot be read. Consider reducing the figure to make it fit on one page (e.g. removing the panel with the bright field only).

Line 297 – 298: Delete ‘the slower insecticide penetration rate, provided by’.

Line 309: I would suggest changing ‘could suggest’ to ‘suggests’.

Materials and methods: This section is technically a bit heavy, which is just the nature of this kind of work. However, it would help the reader if the sections were more structured into paragraphs and sentences summarising the steps at the beginning of a paragraph.

Line 402, ‘Dissected tissues’: Remind the reader what tissues were dissected.

Line 418, ’20 tissues’: I assume you meant to write ’20 tissue samples’ or ’20 samples per tissue type’.

Line 423, ‘[company]’: Is something missing here?

Line 444: Maybe better to write ‘40x magnification lense’ instead of ‘objective’?

Line 486, ‘0.01% 14C-deltamethrin paper’. This needs a bit more explanation here. What method was used to treat the paper? Did it follow the WHO methods, i.e. same type of filter paper? Either provide a reference to the method in the published literature or describe how the papers were prepared.

Line 506: Explain how the dry weight was ‘calculated’.

Line 605: By ‘fork’ do you mean ‘sub company’?

PLOS authors have the option to publish the peer review history of their article (what does this mean?). If published, this will include your full peer review and any attached files.

Reviewer #1: No

Reviewer #2: No
---

## [Editor Report · Decision Letter 1]

28 Jul 2023

Dear Dr Kefi,

We are pleased to inform you that your manuscript 'ABCH2 transporter mediates deltamethrin uptake and toxicity in the malaria vector Anopheles coluzzii' has been provisionally accepted for publication in PLOS Pathogens.

Best regards,

Kenneth D Vernick

Academic Editor

PLOS Pathogens

Margaret Phillips

Section Editor

PLOS Pathogens

Kasturi Haldar

Editor-in-Chief

PLOS Pathogens

orcid.org/0000-0001-5065-158X

Michael Malim

Editor-in-Chief

PLOS Pathogens

orcid.org/0000-0002-7699-2064
---

## [Editor Report · Acceptance letter]

13 Aug 2023

Dear Dr Kefi,

We are delighted to inform you that your manuscript, "ABCH2 transporter mediates deltamethrin uptake and toxicity in the malaria vector Anopheles coluzzii," has been formally accepted for publication in PLOS Pathogens.

Best regards,

Kasturi Haldar

Editor-in-Chief

PLOS Pathogens

orcid.org/0000-0001-5065-158X

Michael Malim

Editor-in-Chief

PLOS Pathogens

orcid.org/0000-0002-7699-2064